# Comparative Study of Gas and Liquid Chromatography Methods for the Determination of Underivatised Neutral and Acidic Cannabinoids and Cholesterol

**DOI:** 10.3390/molecules29102165

**Published:** 2024-05-07

**Authors:** Marian Czauderna, Tomáš Taubner, Wiktoria Wojtak

**Affiliations:** 1The Kielanowski Institute of Animal Physiology and Nutrition, Polish Academy of Sciences, Instytucka 3, 05-110 Jabłonna, Poland; w.wojtak@ifzz.pl; 2Department of Nutrition Physiology and Animal Product Quality, Institute of Animal Science, CZ-104 00 Praha, Czech Republic; taubner.tomas@vuzv.cz

**Keywords:** cannabinoids, cholesterol, gas chromatography, liquid chromatography, mass spectrometry

## Abstract

The aim of our study was to develop a gas chromatographic method coupled with mass spectrometry (GC-MS) for the determination of underivatised neutral (CBDs-N) and acidic (CBDs-A) cannabinoids (CBDs) and cholesterol (Chol). Emphasis was also placed on comparing our original GC-MS method with the currently developed C18-high-performance liquid chromatography with photodiode detection (C18-HPLC-DAD). A combination of a long GC column, shallow temperature column programme, and mass-spectrometry was employed to avoid issues arising from the overlap between CBDs and Chol and background fluctuations. The pre-column procedure for CBDs and Chol in egg yolks consisted of hexane extractions, whereas the pre-column procedure for CBDs in non-animal samples involved methanol and hexane extractions. CBDs-A underwent decarboxylation to CBDs during GC-MS analyses, and pre-column extraction of the processed sample with NaOH solution allowed for CBD-A removal. No losses of CBDs-N were observed in the samples extracted with NaOH solution. GC-MS analyses of the samples before and after extraction with NaOH solution enabled the quantification of CBDs-A and CBDs-N. CBDs-A did not undergo decarboxylation to CBDs-N during C18-HPLC-DAD runs. The use of the C18-HPLC-DAD method allowed simultaneous determination of CBDs-N and CBDs-A. In comparison to the C18-HPLC-DAD method, our GC-MS technique offered improved sensitivity, precision, specificity, and satisfactory separation of underivatised CBDs and Chol from biological materials of endogenous species, especially in hemp and hen egg yolk. The scientific novelty of the present study is the application of the GC-MS method for quantifying underivatised CBDs-A, CBDs-N, and Chol in the samples of interest.

## 1. Introduction

Recent studies have demonstrated that animal products, such as meat, dairy products, or hen eggs are very important dietary sources of nutrients for humans. They provide health-promoting cannabinoids (CBDs), particularly non-psychoactive CBDs, n-3 polyunsaturated fatty acids (n-3PUFA), vitamins, and essential trace elements [1,2,3,4,5]. Fortunately, both diet composition and dietary supplements (e.g., non-psychotropic CBDs and plant or fish oils rich in n-3PUFA) can increase the nutritional value of plant and animal products, thereby benefiting human health [6,7,8,9,10]. Moreover, non-psychotropic CBDs have also shown efficacy in preventing vascular diseases in humans [11]. Interestingly, non-psychotropic CBDs have been found to reduce fungal growth (both in vitro and in vivo), and alter the composition of fungal cell walls and membrane integrity [6]. As a consequence, CBDs can enhance the defence response of fruits, suggesting their potential as a novel, environmentally friendly, post-harvest treatment [6].

CBDs encompass several structural classes of compounds found mainly in cannabis [12]. Interestingly, some CBDs are present in smaller quantities in other plants such as rhododendron, liverworts, North American cone-flower, or flax. Currently, more than 500 types of CBDs have been identified in plants, particularly in cannabis [13]. Recent studies have indicated that oral co-administration of CBDs with oils or high-fat diets result in exceptionally high levels of lipophilic cannabinoids in the intestinal lymphatic system, leading to significant immunomodulatory effects [14]. Due to their lipophilic nature, CBDs efficiently accumulate in lipids and adipose tissues of animals. Therefore, administering CBDs with high-fat diets holds potential as a therapeutic approach to improve the treatment of patients with multiple sclerosis or other autoimmune disorders. Feeding diets enriched in CBDs to livestock may benefit animal welfare by reducing the harmful effects of radicals, such as reactive oxygen and nitrogen species, on the highly responsive cells of their immune system [15]. CBDs (derived from hemp leaves or other parts of this plant) exhibit significant nutritional and potentially therapeutic properties when used as supplements in diets for ruminants, monogastric animals, hens, or humans [3,7]. Recent research has indeed documented the therapeutic effects of CBDs in diseases such as anxiety, epilepsy, as well as motor disorders such as Parkinson’s disease, Alzheimer’s disease, multiple sclerosis, neuropathic pain, schizophrenia, childhood convulsive disorders, or Lennox–Gastaut and Dravet syndromes [5,16]. Low doses of non-psychotropic CBDs (particularly cannabidiol) can be successfully applied as an initial treatment for chronic pain, insomnia, anxiety, stress, and depression [5]. Dietary CBDs have been found to kill cancer cells, reduce tumour growth, relax tight muscles, stimulate appetite, increase bone strength of animals, and improve body weight gain, thereby improving animal welfare [3,17,18,19]. Additionally, ovine diets supplemented with hemp stubble (rich in CBDs) have been shown to positively affect nutrient digestibility [20].

CBDs are classified into three main groups: (a) plant cannabinoids, e.g., cannabis; (b) endocannabinoids—found in animals and humans, e.g., anandamide; and (c) synthetic cannabinoids, such as CP-55940, HU-210, and parahexyl. Interestingly, cannabis isolates (e.g., cannabis oils) contain significant amounts of highly psychoactive compounds, including Δ9-tetrahydrocannabinol (Δ9-THC) and Δ8-tetrahydro-cannabinol (Δ8-THC), which is moderately less active than Δ9-THC [21,22]. The latter (full chemical name: (-)-*trans*-Δ9-tetrahydrocannabinol) is the primary constituent of cannabis responsible for psychoactive effects, whereas other non-psychoactive CBDs do not cause intoxication at typical doses. CBDs can exist in two chemical forms: acidic CBDs (like cannabidiolic acid, cannabidivarinic acid, or cannabichromenic acid) and neutral CBDs (cannabidiol, cannabidivarin, or cannabichromene) [23]. Acidic CBDs (CBDs-A) are the most predominant cannabinoids found in cannabis and hemp plants. As these plants grow, they biosynthesise CBDs-A, which are then converted into the corresponding neutral CBDs (CBDs-N) [12,23,24].

Given the aforementioned points, it is understandable that the use of CBDs, particularly non-psychoactive variants, is gradually becoming accepted. Therefore, it becomes increasingly important to develop sensitive, selective, and straightforward chromatographic methods for quantifying trace concentrations of CBDs in diets, supplements, selected animal and plant tissues, as well as food products [13,25,26,27,28]. Importantly, due to the wide variety of chemical CBDs structures, we intend to determine the concentrations of CBDs-N and CBDs-A in biological samples without pre-column derivatisation of CBDs. We hypothesised that capillary gas chromatography (GC) with mass spectrometry (MS) would provide better selectivity and sensitivity of CBDs analysis than C18-high performance liquid chromatography (C18-HPLC) with photodiode array detection (DAD) (i.e., the C18-HPLC-DAD method).

Considering the above, the main objective of our study was to develop a novel, selective GC with MS (GC-MS) method based on pre-column procedures that would not cause changes in CBDs chemical structures and prevent artefact formation in the analysed samples. Secondly, we aimed to simplify the pre-column procedures for GC-MS determination of CBDs and cholesterol (Chol) in the samples by omitting the derivatisation step. Moreover, the third goal of the present study was to compare the effectiveness of determining underivatised CBDs and Chol by our original GC-MS method with the improved C18-HPLC-DAD method. We expected that, compared to the C18-HPLC-DAD method, the GC-MS technique would enable better sensitivity and separation of CBDs and Chol from endogenous substances.

## 2. Results and Discussion

### 2.1. Determination of Underivatised CBDs Using Gas Chromatography (GC-MS)

The main analytical challenge in the current study was to obtain suitable separation of seventeen CBDs (Table 1) and Chol from the interfering endogenous components present in the biological materials tested. Another issue was obtaining satisfactory separation of CBDs without pre-column derivatisation. To address the problems posed by potential overlap between CBDs, ^GC^IS, Chol, and endogenous species present in biological materials, a low initial column temperature (i.e., 100 °C) was maintained for 1 min (column temperature programme A). Detailed GC-MS analyses of biological materials, particularly hemp and hen egg yolk samples, showed that complete removal of all endogenous components required raising the GC-column temperature to 334 °C (see column temperature programme A; Section 3.3.1). The combination of a long-capillary GC column, shallow temperature column programme A, and selective mass spectrometry provided a suitable analytical tool for simultaneous quantification of CBDs, ^GC^IS, and Chol in hexane solutions, as well as in assayed biological materials. Typical total ion current (TIC) chromatograms for underivatised CBDs, ^GC^IS, and Chol standards (GC-chromatogram A) and the processed hemp biomass sample (GC-chromatogram B) are shown in Figure 1. As anticipated, excellent baseline stability and satisfactory separations of all analytical peaks were achieved, with no observed issues related to overlap of CBDs, ^GC^IS, and Chol peaks in the standard solution (Figure 1A) and biological sample (Figure 1B). Nearly symmetrical CBDs, ^GC^IS, and Chol peak shapes were recorded for the standard samples analysed. Moreover, our original chromatographic method for underivatised CBDs, ^GC^IS, and Chol exhibited good analytical performance within a relatively short analysis time (Figure 1). Unfortunately, GC-MS analyses require high injector and column temperatures, leading to the decarboxylation of the acidic form of CBDs during injections and transit through the GC column [23,28]. In fact, heat or ageing of biological materials containing CBDs-A convert these CBDs acidic forms into neutral forms (i.e., CBDs-A to CBDs-N) (Figure 2).

Considering recent studies [23,28] and our current results, we have argued that some CBDs-N are the sums of the original neutral forms of CBDs (CBDs-N) and CBDs-N formed form decarboxylated CBDs-A during GC-analyses. As a consequence, our TIC-chromatograms documented the presence of CBDs-N peaks (i.e., CBDs-A decarboxylated to CBDs-N; see Figure 1 and Figure 3). The molecular masses (Table 2) and chemical formulas of CBDs-N, Chol, and ^GC^IS in the analysed solutions were confirmed using mass spectra of the standards and the NIST mass spectra library. The similarity of the CBDs-N, ^GC^IS, and Chol peaks in the standard solutions to the mass spectra in the NIST library was over 95%. As expected, the CBDs, ^GC^IS, and Chol peaks were absent in the blank sample when using column temperature programme A and mass selective detection.

### 2.2. GC-MS Analysis of CBDs before and after Pre-Column Extraction with NaOH Solution

The contents of CBDs-A, CBDs-N, ^GC^IS, and Chol in standard solutions and biological materials before and after extraction with aqueous 0.1 M NaOH solution were determined using column temperature programme A with mass-selective detection. Chromatographic analyses indicated that our original extraction with NaOH solution provided satisfactory removal efficiency (~100%) of CBDs-A from standard solutions containing nine CBDs-N and eight CBDs-A (Table 2). The reliability of our GC-method was also assessed using LOD, LOQ, precisions (inter-assay RSD, %), tailing factors (TF) of standard peaks, MS-response to 1 pg of standards, as well as by calculated calibration equations of standards and linear regression coefficients (r) (Table 1). The results demonstrated satisfactory linear MS responses of all standards, directly proportional to their concentrations. In addition, the other validation parameters presented in Table 1 have shown that the pre-column extraction with 0.1 M NaOH solution (instead of CBDs derivatisation [12]), and the original GC-MS method (especially column temperature programme A) are suitable analytical tools for the quantification of CBDs, ^GC^IS, and Chol in standard solutions. Overall, the principal analytical problem addressed in the current study was to develop a satisfactory separation procedure and robust quantitative analysis of CBDs-N and unstable CBDs-A in standard solutions and selected biological materials.

To this end, we analysed the content of CBDs-N and CBDs-A in hexane solutions before and after extraction with 0.1 M NaOH solution (Table 2). The effect of NaOH extraction on CBDs composition in the processed samples is graphically presented in GC-MS chromatograms (Figure 3). As observed in the exemplary GC-MS chromatographic analyses, NaOH extraction completely removed (^CBDs-A^L, % ≈ 100%) CBDs-A in the samples, whereas CBDs-N remained unaffected (^CBDs-N^R, % ≈ 100%). The results presented in Figure 3 and Table 2 and Table 3 demonstrate that GC-MS analysis of the sample containing CBDs-N and CBDs-A before and after extraction with 0.1 M NaOH allows quantification of CBDs-N and CBDs-A in the samples containing both CBDs-N and CBDs-A. Indeed, the concentration of CBDs-N in the assayed solutions could be quantified in the samples extracted with 0.1 M NaOH. On the other hand, an acidic cannabinoid content (see calibration equations in Table 1) could be determined from the difference between the CBDs peak area (^o^S_n_) in the unextracted CBDs solutions and the CBDs peak area (^ext^S_n_) monitored in the solutions extracted using 0.1 M NaOH:S_n_^A^ = ^o^S_n_ − ^ext^S_n_(1)

Moreover, the CBDs-A content in the samples could be determined by n-hexane extraction of aqueous layer A; before CBDs-A extraction, aqueous layer A was acidified with HCl (to pH 1–2). Chromatographic analyses demonstrated that the yield of CBDs-A extraction with hexane was close to 100% (Table 3). On this basis, we proposed that the concentration of CBDs-A could be quantified by extracting with aqueous 0.1 M NaOH solution followed by hexane extraction of the acidified aqueous NaOH solution (i.e., layer A).

The successful development of our original GC-MS method with both pre-column extractions provided the impetus for the application of our novel analytical tools to biological samples. As illustrated in Figure 4, the chromatographic run of the unextracted hemp seed sample (Figure 4A), followed by the chromatographic run of the hemp seed sample extracted with 0.1 M NaOH solution, enabled satisfactory quantification of CBDs-N and CBDs-A in hemp seeds. For a detailed analysis of the identified CBDs peaks in the assayed seeds, the mass spectra of CBDs peaks were corrected by subtracting the background on both sides of the analysed peaks. As anticipated, the corrected mass spectra of CBDs, confirmed the molecular masses and chemical formulas of the detected CBDs in the analysed seeds. This confirmation was achieved by comparing the mass spectra of the CBDs standards, the NIST mass spectra library, and especially the molecular ions (i.e., the highest *m*/*z* value in the mass spectrum of the identified cannabinoid). Moreover, the mass spectra of CBDs in the analysed seeds exhibited a similarity to CBDs standards and/or the mass spectra in the NIST library, exceeding 80%. Similar CBDs profiles, satisfactory separation of CBDs peaks from background fluctuations, and mass spectra similarity of CBDs were found in the egg yolks of hens fed diets enriched with hemp seeds. Moreover, column gradient programme A and MS detection allowed for satisfactory separation of underivatised Chol from background fluctuations and numerous endogenous components in the egg yolk samples. The concentration of Chol in egg yolks ranged from 8.3 to 13.4 mg/g yolk. Dietary fat content tended to correlate positively with Chol concentration in egg yolk samples.

Based on the aforementioned observations, we concluded that our original column gradient programme A and MS detection allowed for satisfactory separation of CBDs and Chol from background fluctuations and endogenous components present in the biological materials tested.

### 2.3. C18-HPLC-DAD Analysis of CBDs and Chol in Standard Solutions and Selected Biological Samples

The ternary gradient elution system consisting of acetonitrile (ACN) with formic acid, water with formic acid, and methanol provided a wide range of solvent strength and excellent baseline stability. However, inadequate separations of CBDs-A, CBDs-N, and Chol, especially in biological samples like egg yolks or hemp seeds, from background fluctuations were observed when two C18 columns were employed. Therefore, three C18 columns filled with a highly hydrophobic silica-based bonded phase was applied to achieve successful separation of acidic and neutral forms of underivatised CBDs and Chol (Table 4; Figure 5). The inclusion of 0.1% formic acid in ACN and water (mobile phases; Table 5) resulted in an improved CBDs peak shape and resolution in chromatographic analysis (Figure 5) compared to other mobile phases without formic acid [28]. Moreover, DAD monitoring at the absorbance maxima of the analysed CBDs was applied to improve the selectivity and sensitivity of CBD detections (Table 4). As expected, all CBDs and Chol peaks were absent from the blank when the ternary gradient elution programme B and DAD monitoring (Table 5) were applied.

In contrast to the GC-MS analysis of CBDs, the HPLC-DAD chromatograms in Figure 5 and results presented in Table 4, demonstrated that the acidic and neutral forms of CBDs were chemically stable (i.e., CBDs-A avoided decarboxylation) during RP-chromatographic analysis. Thus, unlike GC-MS analysis of CBDs-A, C18 chromatography with DAD monitoring permitted simultaneous quantification of CBDs-A and CBDs-N without pre-column extractions with NaOH or hexane of standards solutions or biological samples. Large background fluctuations and presence of endogenous substances in analysed biological materials (especially in egg yolk samples) were the reason for adding 20 µL of the stock ^HPLC^IS solution (i.e., 56 µg ^HPLC^IS) to processed biological samples (see Table 4 and Section 3.1). Moreover, C18 liquid chromatography with DAD monitoring provided a significantly lower DAD response (signals/pg) of CBDs and Chol, and higher LOQ and LOD values than GC with MS detection (Table 1 and Table 4). Thus, our current research confirmed our previous studies, which indicated that GC-MS methods offered better sensitivity, selectivity, and specificity than C18-HPLC-D techniques [12,27,28,29,30,31]. In fact, as shown in Figure 1 and Figure 5, the widths of the CBDs and Chol analytical peaks are smaller in GC-MS chromatograms than in C-18-HPLC-DAD chromatograms. Moreover, background and endogenous components of biological materials interfere more with reliable peak integration in C18-HPLC-DAD analyses compared to capillary GC-MS analyses of CBDs and Chol. Therefore, we argue that our original GC-MS method provided superior sensitivity, precision, and accuracy for the analysis of underivatised CBDs and Chol compared to our improved C18-HPLC-DAD method.

Common methods for the determination of CBDs include GC-MS, GC with flame ionisation detection (GC-FID) and liquid chromatography (HPLC) [12,17,32,33,34,35,36,37,38]. However, the main disadvantage of these methods is the requirement for pre-column derivatisation of acidic CBDs [12,32,37]. On the other hand, there are many interesting previous studies that have utilised modern ultrafast liquid chromatography (e.g., C18-UFLC) and hydrophilic interaction liquid chromatography (HILIC) for efficient separation of acidic and neutral CBDs in biological specimens [17,23,33,34,35,38]. HILIC demonstrated satisfactory selectivity for polar compounds, facilitating excellent separation of CBDs in biological materials. However, HILIC columns typically require longer equilibration times than reversed-phase C18 columns [33,34,35,38]. Fortunately, compared to previously described GC-MS/FID methods [12,32,36,37], our original pre-column protocol with the use of the modern capillary GC-MS method allowed highly sensitive quantification of underivatised acidic and neutral CBDs. Additionally, the utilisation of a 30-m capillary GC-column and mass-selective detection enhanced the separation of over 140 underivatised CBDs in biological samples (such as hemp, egg yolk, muscle, or adipose tissues) compared to currently utilised liquid chromatography approaches (e.g., C18-UFLC or HILIC methods coupled with mass spectrometry). Moreover, an important advantage of GC-MS and GC-FID methods is satisfactory separation of terpenes and neutral CBDs in plant samples (e.g., in hemp samples) and their extracts [32,36].

Our original GC-MS method enables the direct determination of CBDs-N, CBDs-A, and Chol in the samples without pre-column derivatisation of CBDs and Chol compounds. Moreover, pre-column extraction with 0.1 M NaOH solution is significantly simpler compared to pre-column derivatisation of CBDs in biological materials. Pre-column extraction of CBDs-A from processed samples reduces the total content of CBDs compared to GC-MS methods involving pre-column derivatisation of CBDs [12]. Therefore, we believe that the elimination of CBDs derivatisation significantly improves the precision and accuracy of our original GC-MS method compared to GC-MS protocols that include pre-column derivatisation steps.

Considering the above, we recommend our original pre-column procedures and improved GC-MS method (especially our original column temperature programme) for the determination of CBDs-A, CBDs-N, and Chol in animal samples and plant materials, particularly those rich in terpenes or other endogenous components. Unfortunately, Chol levels in plants are very low, which hinders method optimisation in this aspect [39].

In our original GC-MS method, underivatized ^GC^IS (i.e., 5-α-cholestane) served as the internal standard to quantify CBDs and Chol in analysed biological samples. Indeed, previous studies showed that 5-α-cholestane (as the internal standard) has been used to quantify Chol, its oxidation product, steroids, fucosterol, sitosterols, campesterol, and stigmasterol [27,40,41]. Unfortunately, the use of isotopically labelled (e.g., deuterium or ^18^O) Chol or selected CBDs (as internal standards) is very expensive.

Detailed GC-MS and HPLC analyses showed that four extractions with n-hexane and/or methanol (see Section 3.2) allowed for ≥96% CBDs and Chol extraction efficiency from analysed biological samples. The use of ^GC^IS or ^HPLC^IS is particularly important in routine analyses of many biological samples (especially plant and animal tissues rich in endogenous components). In fact, ^GC^IS and ^HPLC^IS determine the effectiveness of separating supernatants (i.e., the solvents used for extractions of CBDs and Chol) from a bottom layer of processed biological samples (i.e., residues derived from analysed biological samples).

## 3. Materials and Methods

### 3.1. Standards and Reagents

Cholesterol (Chol), 5-α-cholestane (internal standard; ^GC^IS), 3,5-dimethylphenol (internal standard; ^HPLC^IS) and all CBDs standards were purchased from Sigma-Aldrich Corp., (St. Louis, MO, USA). The analysed CBDs included: ∆9-tetrahydrocannabinol (∆9-THC), ∆9-tetrahydrocannabinolic acid (∆9-THC-A), ∆8-tetrahydrocannabinol (∆8-THC), cannabidiol (CBD), cannabidiolic acid (CBD-A), cannabinol (CBN), cannabigerol (CBG), cannabigerolic acid (CBG-A), cannabichromene (CBC), cannabichromenic acid (CBC-A), Δ9-tetrahydrocannabivarin (THCV), Δ9-tetrahydrocannabivarinic acid (THCV-A), cannabidivarin (CBDV), cannabidivarinic acid (CBDV-A), cannabicyclol (CBL), and cannabicyclolic acid (CBL-A). The purity of Chol, ^GC^IS, ^HPLC^IS, and all CBDs standards were 99%, ≥97%, ≥99%, and ≥98%, respectively. To prepare stock solutions, Chol, ^GC^IS, and CBDs standards were dissolved in 1 mL of GC-grade n-hexane and stored at −78 °C until use.

GC-grade chloroform (≥99.0%), GC-grade methanol (≥99.8%), ethanol, GC-grade n-hexane (≥99.0%), HPLC-grade acetonitrile (ACN), HPLC-grade methanol, HPLC-grade formic acid, and NaOH were purchased from Merck (Darmstadt, Germany). All other chemicals were of analytical grade and were purchased from Fluka (Steinheim, Germany). Water used to prepare chemical reagents was purified using an Elix™ water purification system (Millipore, Oakville, ON, Canada). High purity He (≥99.9992%) and analytical grade Ar were used, containing 3.7 ppm H_2_O, 1.4 ppm O_2_, 0.1 ppm H_2_, 5.6 ppm N_2_, 0.1 ppm CO, 0.1 ppm CO_2_, and 0.1 ppm alkanes.

Stock solutions of ^GC^IS (160 µg of 5-α-cholestane/mL n-hexane) and ^HPLC^IS (2.8 mg of 3,5-dimethylphenol/mL methanol) were used to calculate the yield of pre-column methods prior to separation on capillary GC and HPLC-C18 columns. To the processed biological samples, 50 µL of ^GC^IS stock solution (i.e., 8 µg of ^GC^IS) or 20 µL of ^HPLC^IS stock solution (i.e., 56 µg of ^HPLC^IS) were added. The final volume of processed biological samples injected onto GC- or HPLC-columns was 1 mL.

### 3.2. Pre-Column Preparation Procedures for Biological Materials

#### 3.2.1. Hen Egg Yolk

The preparation of egg yolk samples consisted of four extractions with n-hexane. To the yolk samples (0.8–1.2 g), 0.8–1.2 mL of water (i.e., 100 µL of water/100 mg of yolk sample), 3.5 mL of n-hexane, and 50 µL of the ^GC^IS solution (prior to GC analyses) or 20 µL of ^HPLC^IS solution (prior to HPLC analyses) were added. All components were protected from light and the resulting mixture was vigorously agitated for 30–40 min using a shaker (800 motion/min). Before removing the upper n-hexane layer, the mixture was centrifuged at 4–5 °C (for ~5 min; 3500 rpm; r = 9 cm), and the supernatant was then transferred to a vial. Subsequently, 3.5 mL of n-hexane was added to the residue, and the extraction procedure with shaking was repeated once more. Afterwards, 3.5 mL of n-hexane was added to the obtained residue. The resulting mixture was ultrasonicated for 30–40 min at ~20 °C. The mixture was centrifuged at 4–5 °C (for ~5 min; 3500 rpm; r = 9 cm), and subsequently, the n-hexane layer was transferred to a vial. Again, 3.5 mL of n-hexane was added to the resulting residue, and the ultrasound extraction was repeated once more. Finally, all upper hexane layers were combined, and hexane was removed under a stream of Ar at ∼40 °C; the residue was stored at −78 °C until further processing. The residue was re-dissolved directly before chromatographic analyses (i.e., GC-MS and HPLC) in 1 mL of GC-grade n-hexane (for GC-MS analyses) or 1 mL of HPLC-grade methanol (for C18-HPLC-DAD analyses). It is recommended to protect the re-dissolved samples from light and store them at −78 °C.

The same pre-column procedure (consisted of four n-hexane extractions) should be applied to homogenised fresh animal tissues (due to the presence of cholesterol).

#### 3.2.2. Plant Materials

Finely powdered samples (~500 mg) of hen feed, hemp seeds, hemp biomass, and flax seeds were mixed with 3.5 mL of methanol and 50 µL of ^GC^IS solution (prior to GC analyses) or 20 µL of ^HPLC^IS solution (prior to HPLC analyses). The resulting mixture was vigorously agitated for 30–40 min using a shaker (800 motion/min). The mixture was centrifuged for ~3 min at ~20 °C (13,000 rpm; r = 3 cm), and the supernatant was collected. Next, 3.5 mL of methanol was added to the obtained residue, and the shaking-extraction procedure was repeated once more. Finally, 3.5 mL of n-hexane was added to the resulting residue. The obtained mixture was ultrasonicated for 30–40 min at ~20 °C. The mixture was centrifuged for ~3 min at ~20 °C (13,000 rpm; r = 3 cm), and the supernatant was collected. Then, 3.5 mL of n-hexane was added to the residue, and ultrasound extraction was repeated once more. Finally, all methanol and n-hexane layers were combined, and the organic solvents were removed under a stream of Ar at ~40 °C and stored at −78 °C until further processing. Directly before chromatographic analyses (i.e., GC-MS and HPLC), the residue was re-dissolved in 1 mL of GC-grade n-hexane (for GC-MS analyses) or 1 mL of HPLC-grade methanol (for C18-HPLC-DAD analyses). It is recommended to protect re-dissolved samples from light and store them at −78 °C.

### 3.3. Chromatographic Equipment and Analytical Methods

#### 3.3.1. Gas Chromatographic Analyses

CBDs and Chol in standard solutions and prepared biological samples were analysed using a “Focus GC” Thermo Scientific gas chromatograph coupled to a Thermo ITQ 1100 mass spectrometer (MS) (Termo Fisher Scientific; Austin, TX, USA). Mass spectra were scanned in the *m*/*z* range from 20 to 500. A Thermo TR-5 ms SQC column (30 m × 0.25 mm i.d. × 0.25 μm film thickness) containing 5% phenyl polysilphenylene-siloxane as the column-phase was used, with He employed as the carrier gas at a constant flow rate of 1.0 mL/min. The injector and transfer line temperatures were maintained at 240 °C and 330 °C, respectively. The MS was operated in the EI mode with full scan monitoring (*m*/*z* 20–500); the ion-source temperature was set to 280 °C (with a limit of 300 °C); electron energy was 70 eV. All CBDs, Chol, and ^GC^IS GC-MS-analyses performed on standard solutions and processed biological samples were based on total ion current (TIC) chromatograms.

CBDs, Chol, and ^GC^IS in the standard solutions and processed biological samples were determined applying column temperature programme A: an initial column temperature of 100 °C was held for 1 min and then increased to 210 °C at a rate of 8 °C/min and held for 7 min before increasing at 1 °C/min to 215 °C and maintained for 7 min; subsequently the temperature was programmed to increase at 2 °C/min to 230 °C, held for 5 min, increased at a rate of 5 °C/min to 250 °C, held for 3 min, increased at 10 °C/min to 300 °C, held for 1 min, increased at 10 °C/min to 320 °C, held for 20 min, increased at 6 °C/min to 334 °C, and held for 10 min. Injections of 1–3 μL of the analysed samples in the splitless injection mode are recommended.

CBDs, Chol, and ^GC^IS identification was validated based on electron impact ionisation spectra of the analysed compounds and compared to authentic CBDs, Chol, and ^GC^IS standards and the NIST 2007 reference mass spectra library. The purity of the CBDs, Chol, and ^GC^IS peaks in standard solutions and biological samples was determined using the NIST MS Search 2.0 mass spectra library. The purity of the CBDs, Chol, and ^GC^IS peaks was assessed based on the similarity of their mass spectra peaks from the samples to those in the NIST mass spectra library. The mass spectra of CBDs, Chol, and ^GC^IS in all biological materials were corrected by subtracting the background on both sides of the CBDs, Chol, and ^GC^IS peaks. The CBDs, Chol, and ^GC^IS peaks in the biological materials analysed were also identified according to the retention time of the separately injected processed standards and by adding these standard solutions to the biological samples.

##### Extraction of Acidic CBDs from Mixtures of CBDs-A and Neutral CBDs-N

One mL of n-hexane solution containing CBDs in standard mixtures or biological samples was vigorously extracted for 1–2 min with one mL of an aqueous 0.1 M NaOH solution. It is strongly recommended to protect the extracted CBDs solutions from light and cool both solutions to ~2 °C just before extraction. The cooled (~2 °C) aqueous 0.1 M NaOH solution used for acidic CBDs extraction should be saturated with n-hexane. After vigorous extraction, the upper hexane layer (supernatant N) was transferred to a GC-vial. Quantification of neutral CBDs in the collected supernatant N was performed using column temperature gradient programme A. The aqueous bottom layer (layer A) containing CBDs-A was stored at −78 °C until further processing.

##### CBDs-A Extraction from Layer A

Layer A was acidified to pH 1–2 with 130 µL of 1 M HCl; 50 µL of ^GC^IS was added to the acidified solution, and then acidic CBDs were extracted for 1–2 min with 950 µL of n-hexane. After vigorous extraction, the upper hexane layer (supernatant A) was transferred to a GC-vial. Quantification of acidic CBDs in the collected supernatant A was performed using column temperature gradient programme A.

#### 3.3.2. Reversed-Phase Liquid-Chromatographic (C18-HPLC-DAD) Analyses

CBDs and Chol in standard solutions and prepared biological samples were determined using a Shimadzu HPLC instrument (VP series) incorporating a liquid chromatograph, an auto-sampler, a communications bus module, a column oven, a degasser, and a selective SPD photodiode array detector (DAD) [31,42]. The analytical columns utilised were Shim-Pak C18 columns (Shimadzu, Kyoto, Japan; mean particle diameter 2.2 µm, column length 75 mm, internal diameter 3 mm) and two Phenomenex Synergy C18 columns (mean particle diameter 2.5 µm, column length 100 mm, internal diameter 3 mm). A guard C18 column (Phenomenex, Torrance, CA, USA; 5 mm × 2 mm) was positioned in front of the analytical columns for their protection. A column heater maintained the temperature at 40 °C. The thermostat of the auto-sampler was set to 4 °C. The DAD was operated in the UV range from 190 to 410 nm. All CBDs in standard solutions and biological samples were analysed using the linear ternary gradient elution programme B of ACN with 0.1% formic acid (*v*/*v*; solvent A), solvent B consisted of water with 0.1% formic acid (*v*/*v*), while solvent C was methanol (Table 5). The maximum system pressure was 29.5 MPa. Injection volumes of biological samples were 5–30 µL. The Chol and CBDs peaks in biological samples were identified based on UV absorption spectra and comparison of retention times with processed Chol and CBDs standards injected separately, and by adding standard solutions to the processed biological samples. Calibration equations were used to determine the concentrations of Chol and CBDs in biological samples (Table 4). To determine CBDs and Chol concentrations in biological samples, ^HPLC^IS was used to quantify the yield of pre-column procedures (Table 4). This yield was determined by assessing the ^HPLC^IS concentrations in the processed biological samples and analysing ^HPLC^IS solutions (56 µg ^HPLC^IS/mL) not subjected to pre-column procedures.

#### 3.3.3. Measurable Assessments of Gas and Liquid Chromatographic Methods

The limit of detection (LOD) was calculated at a signal (S_n_)-to-noise (σ) ratio of 3 (LOD = 3 × S_n_/σ), while the limit of quantification (LOQ) was defined as 10 times the noise under the peak (LOQ = 10 × S_n_/σ); the noise (σ) under the peak was derived from the noise from the left (σ_L_) and right (σ_R_) side of the peak (i.e., σ = (σ_L_ + σ_R_)/2) [42,43,44]. The precision of both chromatographic methods was also assessed by analysing the relative standard deviation (RSD, %) calculated from the measurements of CBDs and Chol concentrations in biological samples as follows [42,45]:RSD, % = (SD/μ) × 100%,
where: SD is the standard deviation of CBDs and Chol measurements in samples, μ is the mean value of CBDs and Chol measurements in standard solutions and biological materials. The precision of the chromatographic methods was determined by analysing the same sample injected multiple times during chromatographic analysis.

The tailing factor (TF) of CBDs-N, CBDs-A, ^GS^IS, ^HPLC^IS, and Chol peaks was calculated as follows:TF = (a + b)/(2 × a),
where: a and b are the peak half-widths at 5% peak height; a is the front half-width; b is the back half-width [29,42].

## 4. Conclusions

The most important scientific novelty of the current study is the utilisation of capillary GC-MS for the quantification of underivatised CBDs-N, CBDs-A, and Chol in biological materials.

Another significant scientific advancement of our study is the use of pre-column extraction with a NaOH solution to remove CBDs-A present in the samples analysed.

Our original GC-MS method can be used for the simple and selective quantification of trace concentrations of underivatised CBDs and Chol in biological materials rich in endogenous components.

We argue that our selective GC-MS method has greater applicability for routine analysis of trace amounts of underivatised CBDs-N, CBDs-A, or/and Chol in biological samples compared to liquid chromatography such as reversed-phase HPLC/UFLC or HILIC.

For future studies, we suggest the implementation of pre-column extraction with NaOH solution, longer capillary GC columns (e.g., 60 m) and GC-MS/MS techniques for the determination of underivatised CBDs, Chol, and oxidised Chol (Ox-Chol) in animal tissues (especially Chol-rich) and underivatised CBDs and Chol in plant materials like hemp). These new analytical techniques can improve selectivity, sensitivity, accuracy, and precision of CBDs, Chol,21 and Ox-Chol quantification in biological samples (especially in animal samples).

## Figures and Tables

**Figure 1 molecules-29-02165-f001:**
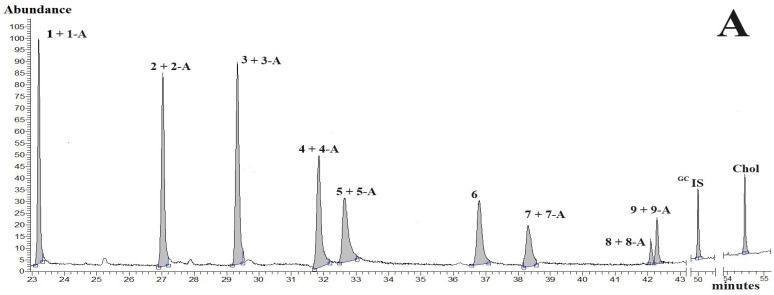
Typical GC-MS chromatograms using column temperature programme A. (**A**) (TIC-chromatogram A)—the chromatogram for nine CBDs-N, eight CBDs-A, ^GC^IS (internal standard; 5-α-cholestane) and Chol. (**B**) (TIC-chromatogram B)—the chromatogram for the processed hemp biomass sample. Peaks: 1—CBDV and CBDV from decarboxylation of CBDV-A (1-A); 2—THCV and THCV from decarboxylation of THCV-A (2-A); 3—CBL and CBL from decarboxylation of CBL-A (3-A); 4—CBD and CBD from decarboxylation of CBD-A (4-A); 5—CBC and CBC from decarboxylation of CBC-A (5-A); 6—∆8-THC; 7—∆9-THC and ∆9-THC from decarboxylation of ∆9-THC-A (7-A); 8—CBG and CBG from decarboxylation of CBG-A (8-A); 9—CBN and CBN from decarboxylation of CBN-A (9-A). (**B**)—Chol concentration in the hemp biomass sample was below the limit of detection (LOD). Blue lines under peaks—baselines under integrated peaks; **□−−□**—an integration range of peaks.

**Figure 2 molecules-29-02165-f002:**
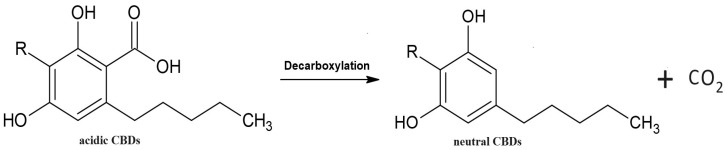
Schematic of the decarboxylation reaction of the acidic form of CBDs (CBDs-A) to neutral cannabinoids (CBDs-N).

**Figure 3 molecules-29-02165-f003:**
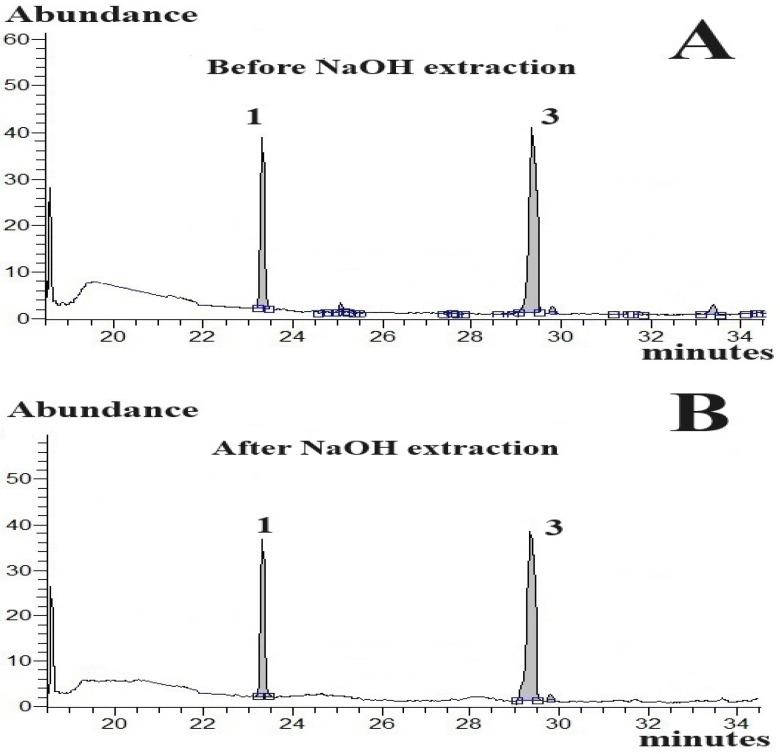
Effect of extraction with aqueous 0.1 M NaOH solution on the composition of CBDs hexane solutions. Parts of typical GC-MS chromatographic runs: (**A**) analysed CBDs-N, ^GC^IS chromatographic analysis of unextracted neutral CBDs (i.e., CBDV and CBL); (**B**) chromatographic analysis of neutral CBDs (i.e., CBDV and CBL) extracted with 0.1 M NaOH; (**C**) chromatographic analysis of unextracted acidic CBDs (i.e., CBD-A and CBC-A); (**D**) chromatographic analysis of acidic CBDs (i.e., CBD-A and CBC-A) extracted with 0.1 M NaOH solution. Peaks: 1—CBDV; 3—CBL; 4-A—CBD-A; 5-A—CBC-A. Blue lines under peaks—baselines under integrated peaks; **□−−□**—an integration range of peaks.

**Figure 4 molecules-29-02165-f004:**
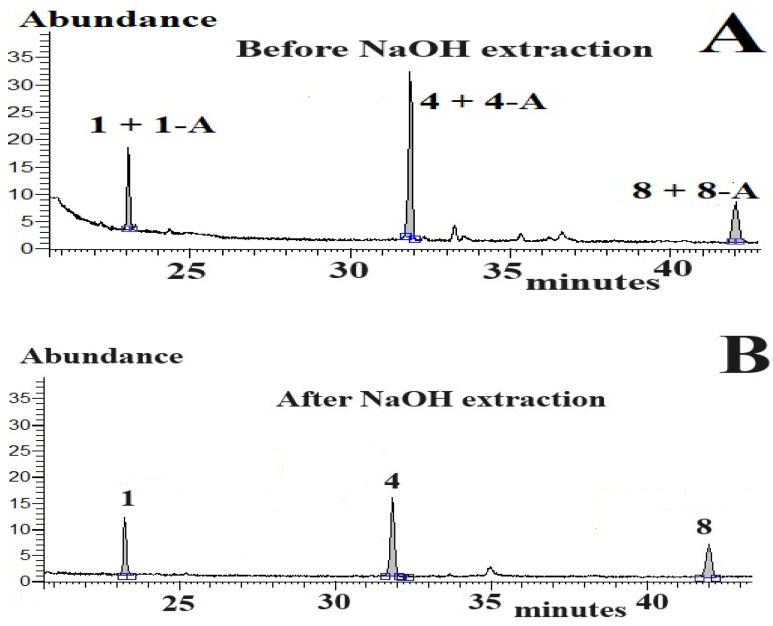
CBDs concentrations in processed hemp seeds (~0.5 g) using the GC-MS method and column programme A (injection volume: 1 µL). Parts of GC-MS chromatograms of hemp seed samples before (**A**) extraction with 0.1 M NaOH solution and after extraction (**B**) with 0.1 M NaOH solution. Concentration of detected CBDs-N: peak 1: CBDV—3.37 µg/g; peak 4: CBD—6.06 µg/g; peak 8: CBG—3.11 µg/g. Concentration of detected CBDs-A: peak 1-A: CBDV-A—1.32 µg/g; peak 4-A: CBD-A—3.39 µg/g; peak 8-A: CBG-A—1.01 µg/g. Blue lines under peaks—baselines under integrated peaks; **□−−□**—an integration range of peaks.

**Figure 5 molecules-29-02165-f005:**
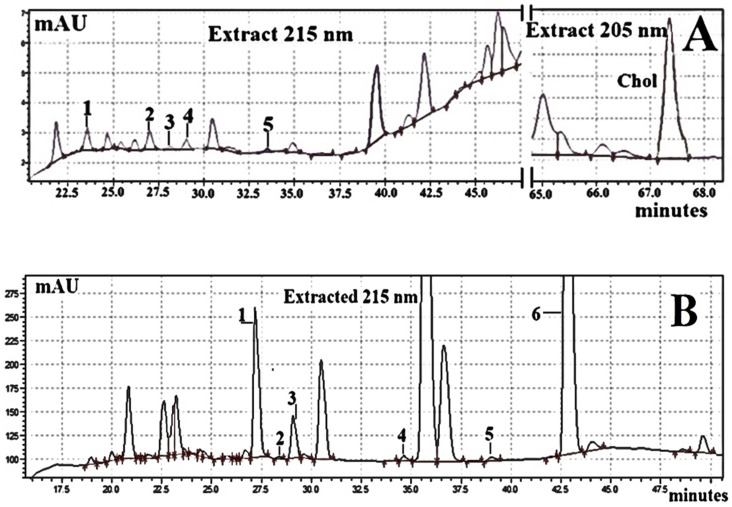
Fragments of the C18-HPLC-DAD chromatogram of the processed biological samples (~500 mg). (**A**): chromatogram A—egg yolk of hens fed a diet enriched with hemp seeds. Peak 1—CBDV; peak 2—CBD-A; peak 3—CBG-A; peak 4—CBD; peak 5—THCV-A. (**B**): chromatogram B—hemp seeds. Peak 1—CBD-A; peak 2—CBG-A; peak 3—CBD; peak 4—CBN; peak 5—∆9-THC; peak 6—CBC; Chol concentration in processed hemp seeds was below LOD. ↓ and ↑—integration ranges of peaks.

**Table 1 molecules-29-02165-t001:** Retention times of analysed CBDs, ^GC^IS (as 5-α-cholestane), and Chol standards, calibration equations, linearity (correlation coefficients; r), detection (LOD), quantification (LOQ) limits, the inter-assay (RSD, %) precision and peak tailing factors (TF) of assayed standards determined in standard solutions using GC-MS method and column temperature program A (injection volumes: 1–3 µL).

Compound	Retention Time, min (Mean ± SD)	^a o^S_n_ before Extraction	^b ext^S_n_ after Extraction	^c^ Recovery (R), %	^d^ Calibration Equations y(ng) = a × S_n_	^e^ r; Linear Regression Coefficient	^f^ LOD pg/mL	^f^ LOQ pg/mL	^g^ Inter- Assay RSD, %	^h^ Tailing Factor (TF) of a Peak	^i^ MS Response to 1 pg of Standards
Neutral forms of cannabinoids (CBDs-N)											
CBDV	23.2 ± 0.1	576,784	577,484	100.1	y = 2.68 × 10^−6^ × S_n_	0.9968	6.8 (0.024)	22.7 (0.08)	1.21	0.994	373
THCV	27.1 ± 0.1	623,956	622,793	99.8	y = 2.40 × 10^−6^ × S_n_	0.9968	5.9 (0.021)	19.4 (0.07)	1.32	0.990	417
CBL	29.3 ± 0.1	539,796	540,184	100.1	y = 2.05 × 10^−6^ × S_n_	0.9973	2.8 (0.009)	9.3 (0.03)	1.09	0.991	487
CBD	31.9 ± 0.1	414,490	413,206	99.7	y = 2.59 × 10^−6^ × S_n_	0.9976	5.0 (0.015)	16.8 (0.05)	1.46	0.987	387
CBC	32.6 ± 0.1	298,395	297,171	99.6	y = 2.34 × 10^−6^ × S_n_	0.9974	5.8 (0.019)	19.4 (0.06)	1.54	0.993	427
∆8-THC	36.8 ± 0.1	324,754	323,256	99.5	y = 1.91 × 10^−6^ × S_n_	0.9982	6.1 (0.022)	20.3 (0.07)	1.07	0.987	523
∆9-THC	38.3 ± 0.1	248,395	250,171	100.8	y = 2.60 × 10^−6^ × S_n_	0.9969	1.6 (0.007)	5.4 (0.02)	1.29	0.986	385
CBG	42.1 ± 0.1	154,899	155,901	100.6	y = 2.69 × 10^−6^ × S_n_	0.9981	4.0 (0.013)	13.2 (0.04)	2.23	0.988	372
CBN	42.4 ± 0.1	356,274	353,893	99.3	y = 5.89 × 10^−6^ × S_n_	0.9990	5.5 (0.018)	18.2 (0.06)	2.07	0.990	170
Acidic forms of cannabinoids (CBDs-A)											
CBDV-A ^j^ (CBDV)	- 23.2 ± 0.1 ^k^	- 595,039 ^k^	- 0	0 0	y = 2.87 × 10^−6^ × S_n_ -	0.9987 -	7.3 (0.022) -	24.3 (0.07) -	1.18 -	0.993 -	348 -
THCV-A ^j^ (THCV)	- 27.1 ± 0.1 ^k^	- 607,956 ^k^	- 0	0 0	y = 2.76 × 10^−6^ × S_n_ -	0.9963 -	6.8 (0.022) -	22.6 (0.07) -	1.28 -	0.991 -	461 -
CBL-A ^j^ (CBL)	- 29.3 ± 0.1 ^k^	- 576,831 ^k^	- 0	0 0	y = 2.09 × 10^−6^ × S_n_ -	0.9980 -	2.9 (0.009) -	9.5 (0.03) -	1.13 -	0.990 -	479 -
CBD-A ^j^ (CBD)	- 31.9 ± 0.1 ^k^	- 442,352 ^k^	- 0	0 0	y = 2.95 × 10^−6^ × S_n_ -	0.9984 -	5.7 (0.0015) -	19.1 (0.05) -	1.43 -	0.988 -	339 -
CBC-A ^j^ (CBC)	- 32.6 ± 0.1 ^k^	- 276,873 ^k^	- 0	0 0	y = 2.67 × 10^−6^ × S_n_ -	0.9976 -	6.6 (0.019) -	22.1 (0.06) -	1.52 -	0.992 -	375 -
∆9-THC-A ^j^ (∆9-THC)	- 38.3 ± 0.1 ^k^	- 496,482 ^k^	- 0	0 0	y = 2.96 × 10^−6^ × S_n_ -	0.9979 -	1.8 (0.009) -	6.1 (0.02) -	1.12 -	0.987 -	338 -
CBG-A ^j^ (CBG)	- 42.1 ± 0.1 ^k^	- 184,328 ^k^	- 0	0 0	y = 3.07 × 10^−6^ × S_n_ -	0.9983 -	4.5 (0.013) -	15.0 (0.04) -	2.18 -	0.989 -	326 -
CBN-A ^j^ (CBN)	- 42.4 ± 0.1 ^k^	- 395,373 ^k^	- 0	0 0	y = 6.71 × 10^−6^ × S_n_ -	0.9987 -	6.2 (0.019) -	20.7 (0.06) -	1.93 -	0.991 -	149 -
^GC^IS ^l^	50.6 ± 0.1	299,184 ^m^	297,705	99.5	y = 2.669 × 10^−5^ × S_n_	0.9991	62.8 (0.17)	209.0 (0.56)	1.32	0.992	37
Chol	54.5 ± 0.2	623,523	621,097	99.6	y = 8.37 × 10^−7^ × S_n_	0.9944	18.2 (0.05)	60.5 (0.16)	1.49	0.989	1094

SD—standard deviation; RSD—relative standard deviation; CBDs—cannabinoids; Chol—cholesterol; ^GC^IS—internal standard used for calculation of the pre-column extraction yield of CBDs and Chol; S_n—_peak area. ^a o^S_n_ (injection volume: 1 µL)—CBDs, ^GC^IS and Chol peak areas monitored in standard solutions before extraction with aqueous 0.1 M NaOH solution. ^b ext^S_n_ (injection volume: 1 µL)—CBDs, ^GC^IS and Chol peak areas monitored in standards solutions after extraction with 0.1 M NaOH. ^c^ Recovery (R) was calculated as follows: R, % = (^ext^S_n_/^o^S_n_) × 100%. ^d^ S_n_—compound peak areas obtained by MS detection; linear regression forcing the intercept on point 0,0; number of points used in the calibration curves: 5 (i.e., five sets of concentrations of CBDs, ^GC^IS and Chol standards were used for preparing the calibration curves); the amount ranges of injected CBDs and Chol standards: 2–30 and 3–39 ng/injection, respectively. ^e^ Numerical measure of the statistical relationship between MS responses and quantities of injected standards. ^f^ LOD and LOQ values are given in parentheses in pmol/mL. ^g^ Precision (RSD, %) was evaluated by multianalysis of the assayed standards (CBDs-N, CBDs-A, ^GC^IS and Chol). ^h^ Tailing factor (TF) [29] of CBDs, ^GC^IS and Chol peaks for each standard were calculated based on five concentrations of CBDs-N, CBDs-A, ^GC^IS and Chol standards used for preparing the calibration equations. ^i^ Recorded number of detector (MS) signals per 1 pg of analysed CBDs or Chol. ^j^ Acidic forms of CBDs (CBDs-A) are decarboxylated to the appropriate neutral form of CBDs (CBDs-N); see Figure 2. CBDs-N formed from the decarboxylated CBDs-A are given in parentheses. ^k^ Retention times and peak areas (S_n_) of neutral CBDs—products (in brackets) of decarboxylation of acidic CBDs (i.e., the substrate of decarboxylation reaction). ^l^ Internal standard: 8 µg ^GC^IS were added to analysed biological samples; the final volume of these samples: 1 mL. The amount range of ^GC^IS in 1 mL of calibration ^GC^IS solutions: 2–10 µg ^GC^IS (the injection volumes: 1 µL; i.e., the amount range of injected ^GC^IS onto capillary GC column: 2–10 ng/injection). ^m^ The injection volume of the calibration ^GC^IS solution (8 µg ^GC^IS/mL; 8 ng/injection): 1 µL (4—number of replicates).

**Table 2 molecules-29-02165-t002:** Recoveries ^a^ (^CBDs-N^R, % ^b^ and ^CBDs-A^R, % ^c^) of CBDs-N and CBDs-A and loss (^CBDs-A^L, %) ^d^ of CBDs-A contents in standards solutions extracted with NaOH solution using the GC-MS method and column temperature programme A (3 replicates; injection volumes: 1, 2, or 3 µL).

Cannabinoid in Analysed Solutions	MM g/mol	Retention Time, min (Mean ± SD)	Recoveries of CBDs after Extraction with 0.1 M NaOH Solution		^CBDs-A^L, % in Solutions Containing CBDs-N and CBDs-A
			^CBDs-N^R, % in CBDs-N Solutions	^CBDs-A^R, % in CBDs-A Solutions	
CBDV ^e^	286.4	23.2 ± 0.1	99 ± 1	- ^g^	- ^h^
CBDV-A ^f^	330.4	23.2 ± 0.1	- ^g^	0	101 ± 2
THCV ^e^	286.4	27.1 ± 0.1	101 ± 1	- ^g^	- ^h^
THCV-A ^f^	330.4	27.1 ± 0.1	- ^g^	0	99 ± 1
CBL ^e^	314.5	29.3 ± 0.1	98 ± 1	- ^g^	- ^h^
CBL-A ^f^	358.5	29.3 ± 0.1	- ^g^	0	98 ± 2
CBD ^e^	314.5	31.9 ± 0.1	100 ± 1	- ^g^	- ^h^
CBD-A ^f^	358.5	31.9 ± 0.1	- ^g^	0	101 ± 1
CBC ^e^	314.5	32.6 ± 0.1	98 ± 1	- ^g^	- ^h^
CBC-A ^f^	358.5	32.6 ± 0.1	- ^g^	0	99 ± 2
∆8-THC	314.5	36.8 ± 0.1	101 ± 1	- ^g^	- ^h^
∆9-THC ^e^	314.5	38.3 ± 0.1	99 ± 2	- ^g^	- ^h^
∆9-THC-A ^f^	358.5	38.3 ± 0.1	- ^g^	0	101 ± 2
CBG ^e^	316.5	42.1 ± 0.1	99 ± 1	- ^g^	- ^h^
CBG-A ^f^	360.5	42.1 ± 0.1	- ^g^	0	99 ± 1
CBN ^e^	310.5	42.4 ± 0.1	102 ± 2	- ^g^	- ^h^
CBN-A ^f^	354.5	42.4 ± 0.1	- ^g^	0	101 ± 1

MM—molar mass of assayed CBDs. ^a^ Recovery (R, %) was calculated as follows: R, % = (^ext^S_n_/^o^S_n_) × 100%, where: ^ext^S_n—_CBDs-peak area monitored in solutions extracted with 0.1 M NaOH solution; ^o^S_n_—CBDs-peak area monitored in solutions before extraction with 0.1 M NaOH solution. ^b CBDs-N^R, %—recovery for neutral CBDs (CBDs-N). ^c CBDs-A^R, %—the recovery for acidic CBDs (CBDs-A); injected quantities of CBDs-N and CBDs-A: from 10 to 30 ng. ^d^ Loss of CBDs-A content in CBDs solutions extracted with 0.1 M NaOH: ^CBDs-A^L, % = ((^o^S_n_^N^ + ^o^S_n_^A^ − ^ext^S_n_)/^o^S_n_^A^) × 100%, where: ^o^S_n_^N^—CBDs-N peak area monitored in CBDs-N solutions before extraction with 0.1 M NaOH; ^o^S_n_^A^—CBDs-A peak area monitored in CBDs-A solutions before extraction with 0.1 M NaOH; ^ext^S_n_—cannabinoid peak area monitored in extracted solutions containing both neutral and acidic form of cannabinoid; concentration of both CBDs forms in all solutions was 10 µg/mL.^e^ Neutral CBDs; these neutral CBDs forms are formed from decarboxylation of CBDs-A. ^f^ Acidic CBDs. ^g^ Not present in assayed solutions. ^h^ No loss of assayed CBDs-N; recoveries (R, %) of CBDs-N in solutions containing acidic and neutral CBDs were 99 ± 2%.

**Table 3 molecules-29-02165-t003:** Recoveries (^CBDs^R, % and ^CBDs^Re, %) for exemplary neutral and acidic CBDs in standard solutions using the GC-MS method and column temperature programme A.

Cannabinoid in Assayed Solutions	^o^ S_n_ ^a^ before Extraction with 0.1 M NaOH	^CBDs^R, % ^b^ in Solutions Extracted with 0.1 M NaOH	^CBDs^Re, % ^c^ in Solutions Re-Extracted with Hexane
CBDV ^d^	288,392	100.1	0
CBL ^d^	268,964	100.1	0
CBD-A ^e^	221,984	0	98.7
CBC-A ^e^	137,492	0	98.9

^a o^S_n_—CBDs peak areas monitored in standard solutions before extraction with aqueous 0.1 M NaOH solution. ^b^ Recovery (^CBDs^R, %) was calculated as follows: ^CBDs^R, % = (^ext^S_n_/^o^S_n_) × 100%, where: ^ext^S_n—_CBDs peak areas monitored in standard solutions after extraction with 0.1 M NaOH. ^c CBDs^Re, %—recovery of CBDs-A for hexane extraction of aqueous NaOH solutions (layer A) acidified with 1 M HCl (to pH 1–2). ^d^ Neutral cannabinoids (CBDs-N). ^e^ Acidic cannabinoids.

**Table 4 molecules-29-02165-t004:** Retention times of analysed CBDs, Chol standards, ^HPLC^IS ^a^, calibration equations, linearity (correlation coefficients; r), detection (LOD), and quantification (LOQ) limits and tailing factors (TF) of assayed standards determined in standard solutions using the C18-HPLC-DAD method and ternate gradient elution programme B.

Compound	Retention Time, min (Mean ± SD)	DAD Detection (nm)	Calibration Equations ^b^ y(µg) = a × S_n_	r; Linear Regression Coefficient ^c^	LOD ^d^ ng/mL	LOQ ^d^ ng/mL	Tailing Factor (TF) of a Peak ^e^	DAD Response to 1 pg of Standards
^HPLC^IS	9.25 ± 0.05	206	y = 5.861 × 10^−6^ × S_n_	0.9994	9.6 (79)	32.0 (263)	0.999	0.171
CBDV-A ^f^	22.9 ± 0.1	221	y = 8.165 × 10^−8^ × S_n_	0.9999	20.6 (62)	60.8 (208)	0.998	12.2
		268	y = 1.727 × 10^−7^ × S_n_	0.9998	3.3 (10)	10.9 (33)	0.997	5.8
CBDV ^g^	23.7 ± 0.1	205	y = 4.467 × 10^−8^ × S_n_	0.9999	14.7 (51)	49.1 (172)	0.998	22.4
		273	y = 1.767 × 10^−6^ × S_n_	0.9998	9.8 (34)	32.8 (115)	0.997	0.6
CBD-A ^f^	27.1 ± 0.1	221	y = 8.778 × 10^−8^ × S_n_	0.9999	26.0 (73)	86.8 (242)	0.996	11.4
		268	y = 1.876 × 10^−7^ × S_n_	0.9999	4.2 (12)	13.9 (39)	0.996	5.3
CBG-A ^f^	28.1 ± 0.1	220	y = 8.168 × 10^−8^ × S_n_	0.9999	27.4 (76)	91.5 (254)	0.998	12.2
		268	y = 1.733 × 10^−7^ × S_n_	0.9999	4.2 (12)	13.9 (39)	0.999	5.8
CBG ^g^	28.5 ± 0.1	205	y = 4.762 × 10^−8^ × S_n_	0.9999	18.9 (60)	63.1 (199)	0.996	21.0
		273	y = 2.137 × 10^−6^ × S_n_	0.9999	14.2 (45)	47.5 (150)	0.997	0.5
CBD ^g^	29.1 ± 0.1	205	y = 4.617 × 10^−8^ × S_n_	0.9999	19.5 (62)	65.1 (207)	0.998	21.7
		274	y = 1.839 × 10^−6^ × S_n_	0.9999	13.8 (44)	46.1 (146)	0.999	0.54
THCV ^g^	29.4 ± 0.1	205	y = 5.774 × 10^−8^ × S_n_	0.9999	23.3 (81)	77.7 (271)	0.995	17.3
		278	y = 2.139 × 10^−6^ × S_n_	0.9999	15.4 (54)	51.4 (180)	0.996	0.47
THCV-A ^f^	33.7 ± 0.1	220	y = 1.007 × 10^−7^ × S_n_	0.9999	45.2 (137)	150.8 (457)	0.996	9.9
		269	y = 1.779 × 10^−7^ × S_n_	0.9999	5.3 (16)	17.7 (53)	0.997	5.6
CBN ^g^	34.6 ± 0.1	215	y = 6.793 × 10^−8^ × S_n_	0.9999	37.8 (122)	125.9 (405)	0.997	14.7
		284	y = 1.247 × 10^−7^ × S_n_	0.9999	1.3 (4.2)	4.3 (14)	0.996	8.0
∆9-THC ^g^	39.0 ± 0.1	209	y = 6.343 × 10^−8^ × S_n_	0.9999	34.7 (110)	115.6 (368)	0.996	15.8
		279	y = 2.492 × 10^−6^ × S_n_	0.9965	21.4 (68)	71.2 (224)	0.996	0.4
∆8-THC ^g^	39.8 ± 0.1	207	y = 7.548 × 10^−8^ × S_n_	0.9999	50.9 (162)	169.8 (540)	0.995	13.2
		279	y = 2.833 × 10^−7^ × S_n_	0.9999	35.8 (113)	119.3 (372)	0.995	3.5
CBN-A ^f^	40.3 ± 0.1	262	y = 7.270 × 10^−8^ × S_n_	0.9999	0.4 (1.2)	1.4 (4.1)	0.996	13.8
		326	y = 3.954 × 10^−7^ × S_n_	0.9999	2.2 (6.6)	7.3 (22)	0.996	2.5
CBL ^g^	42.0 ± 0.2	210	y = 6.044 × 10^−8^ × S_n_	0.9999	24.7 (78)	82.2 (261)	0.995	16.5
		278	y = 1.743 × 10^−6^ × S_n_	0.9998	10.2 (32)	33.8 (106)	0.995	0.6
CBC ^g^	43.0 ± 0.1	230	y = 1.100 ×10^−7^ × S_n_	0.9999	17.0 (54)	56.6 (180)	0.997	9.1
		279	y = 2.723 × 10^−7^ × S_n_	0.9999	2.0 (6.3)	6.6 (21)	0.996	3.7
∆9-THC-A ^f^	43.9 ± 0.1	220	y = 1.077 × 10^−7^ × S_n_	0.9999	40.8 (114)	136.2 (380)	0.996	9.3
		271	y = 1.927 × 10^−7^ × S_n_	0.9999	4.2 (12)	14.1 (39)	0.995	5.2
CBC-A ^f^	46.3 ± 0.1	199	y = 1.394 × 10^−7^ × S_n_	0.9996	36.1 (101)	120.3 (336)	0.995	7.2
		254	y = 8.045 × 10^−8^ × S_n_	0.9997	0.4 (1.2)	1.5 (4.0)	0.994	12.4
CBL-A ^f^	46.6 ± 0.1	227	y = 1.969 × 10^−7^ × S_n_	0.9999	43.5 (122)	145.1 (405)	0.994	5.1
		273	y = 3.472 × 10^−7^ × S_n_	0.9997	2.8 (7.9)	9.4 (26)	0.993	2.9
Chol	67.2 ± 0.3	205	y = 2.240 × 10^−7^ × S_n_	0.9902	144.4 (373)	480.8 (1244)	0.991	4.5

^a^ Internal standard (as 3,5-dimethylphenol): 56 µg ^HPLC^IS were added to analysed biological samples; the final volume of these samples: 1 mL. The amount range of ^HPLC^IS: 20–60 µg ^HPLC^IS in 1 mL of calibration ^HPLC^IS solutions (amount range of injected ^HPLC^IS onto C18 columns: 0.4–1.2 μg/injection; injection volume: 20 µL). ^b^ S_n_—compound peak areas obtained using DAD monitoring; linear regression forcing intercept 0,0; number of points used in the calibration curves: 6 (i.e., six sets of concentrations of ^HPLC^IS, CBDs, and Chol standards were used to prepare the calibration curves); amount range of injected CBDs and Chol standards dissolved in methanol: 0.05–0.40 μg/injection. ^c^ Numerical measure of the statistical relationship between DAD responses and amounts of injected standards. ^d^ LOD and LOQ values in parentheses are given in pmol/mL. ^e^ TF of CBDs, ^HPLC^IS, and Chol peaks [29]. ^f^ Acidic CBDs (CBDS-A). ^g^ Neutral CBDs (CBDs-N).

**Table 5 molecules-29-02165-t005:** Ternary gradient elution programme B and DAD monitoring used for C18-HPLC-DAD analysis of CBDs in standard solutions and biological samples.

Time, min	Flow-Rate min/mL	Composition, % ^a^		
		Solvent A	Solvent B	Solvent C
0	0.30	55	45	0
2.5	0.30	55	45	0
4.0	0.30	70	30	0
9.0	0.30	75	25	0
17.0	0.30	75	25	0
19.0	0.35	75	25	0
26.0	0.35	75	25	0
30.0	0.35	82	18	0
32.0	0.40	85	15	0
35.0	0.40	85	15	0
39.0	0.40	92	8	0
43.0	0.40	99	1	0
44.0	0.40	100	0	0
46.2	0.40	0	0	100
70.0 ^b^	0.41	0	0	100

^a^ Solvent A—ACN (HPLC-grade acetonitrile) with 0.1% formic acid (*v*/*v*); solvent B—water with 0.1% formic acid (*v*/*v*); solvent C—HPLC-grade methanol. ^b^ After 70 min, the C18 columns were re-equilibrated for 18 min in 55% solvent A and 45% solvent B at a flow-rate of 0.3 mL/min.

## Data Availability

Dataset available on request from the authors.

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
