# Peer review of "Comparative Study of Gas and Liquid Chromatography Methods for the Determination of Underivatised Neutral and Acidic Cannabinoids and Cholesterol"

_molecules, 2024, doi:10.3390/molecules29102165_

Round 1

Reviewer 1 Report

Comments and Suggestions for Authors

The point is good enough for acceptance with no observation modifications. The novelty is also good and the expermental desing is well considred. I do recommend this manuscripit to be accepted. 

Author Response

Reviewer #1

We would like to thank Reviewer #1 for his opinion

Reviewer 2 Report

Comments and Suggestions for Authors

The prepared manuscript number "molecules-2942864" "Comparative study of gas and liquid chromatography methods for determining neutral and acidic cannabinoids and cholesterol ". This study introduces a novel approach using GC-MS for quantifying underivatized neutral and acidic cannabinoids and cholesterol in samples. I have some remarks before they can be published.

1-      The English of the entire manuscript should be significantly revised from a professional to be suitable for publication.

2-      The introduction section should have more recent data about the cannabinoids.

3-      At the end of the introduction section, the objective of the work should be more clearly stated.

4-      The data presented hides the amount of work performed. It is noticeable that the work is good and has large results, but the discussion was too descriptive and lacked the explanation of the phenomenon they observed, the authors should discuss the results more deeply with recent references (2021-2024).

5-      Improve tables quality and charts.

6-      conclusions should be rewritten not to repeat results but to highlight most important ones and conclude main aspects of the work, implications, and future prospects.

7-      In the reference section the style of some references should be corrected. The Journal names should all either abbreviated or their full names provided.

Comments on the Quality of English Language

The prepared manuscript number "molecules-2942864" "Comparative study of gas and liquid chromatography methods for determining neutral and acidic cannabinoids and cholesterol ". This study introduces a novel approach using GC-MS for quantifying underivatized neutral and acidic cannabinoids and cholesterol in samples. I have some remarks before they can be published.

1-      The English of the entire manuscript should be significantly revised from a professional to be suitable for publication.

2-      The introduction section should have more recent data about the cannabinoids.

3-      At the end of the introduction section, the objective of the work should be more clearly stated.

4-      The data presented hides the amount of work performed. It is noticeable that the work is good and has large results, but the discussion was too descriptive and lacked the explanation of the phenomenon they observed, the authors should discuss the results more deeply with recent references (2021-2024).

5-      Improve tables quality and charts.

6-      conclusions should be rewritten not to repeat results but to highlight most important ones and conclude main aspects of the work, implications, and future prospects.

7-      In the reference section the style of some references should be corrected. The Journal names should all either abbreviated or their full names provided.

Author Response

1-     The English of the entire manuscript should be significantly revised from a professional to be suitable for publication.

Our responseOur revised manuscript was thoroughly checked by the language expert and native speaker. We do hope that this allow to avoid both language and grammar mistakes in present manuscript.

2-      The introduction section should have more recent data about the cannabinoids.

Our responseWe added more recent data about the cannabinoids:

                          see lines: 40-45, 64-66;

                          see references: [2], [5], [6], [11], [33] and [34].

3-      At the end of the introduction section, the objective of the work should be more clearly stated.

Our response:   The goals of our work have been more clearly defined:

                            see lines:  95-104.

4-      The data presented hides the amount of work performed. It is noticeable that the work is good and has large results, but the discussion was too descriptive and lacked the explanation of the phenomenon they observed, the authors should discuss the results more deeply with recent references (2021-2024).

Our response:      The interpretation of the results obtained has been improved:

                               see lines:  113-122, 326-368 and 533-549. 

                               Moreover, the recent references (2021-2024) were added: [2], [5], [6], [11], [33], [34], [37] and [38].

5-      Improve tables quality and charts.

      Our response:        We improved the readability and quality of Tables and Figures.

6-      Conclusions should be rewritten not to repeat results but to highlight most important ones and conclude main aspects of the work, implications, and future prospects.

      Our response:    The conclusion has been improved:

                                 See lines:  533-549.

7-      In the reference section the style of some references should be corrected. The Journal names should all either abbreviated or their full names provided.

               Our response:  In the reference section, the style of references has been corrected.  The Journal names have been abbreviated. 

Reviewer 3 Report

Comments and Suggestions for Authors

Dear Authors,

the work entitle "Comparative study of gas and liquid chromatography methods for determining neutral and acidic cannabinoids and cholesterol" was well conducted and presented some new pre sample treatment originality for acid CBDs.

The work was well written, but in my opinion the discussion could be improved.

The scientific novelty of the study is specially focused on use of capillary GC-MS for quantification of underivatized CBDs and when  compared to C18-HPLC-DAD method. The new method proved to be useful. However,  in my opinion, the discussion could be improved by including some interesting data from other researches who were well successful on separation of underivatized metabolites from Cannabis using different HPLC methods in one unique step process on contrary to this new CG-MS pre-extraction method. There's many interesting work that also used RP-HPLC and HILIC columns to efficiently separate acid CDBs in fast way. So, it would be interesting to talk with those data in the discussion since it seems to be very practical method too.

Author Response

  The scientific novelty of the study is specially focused on use of capillary GC-MS for quantification of underivatized CBDs and when  compared to C18-HPLC-DAD method. The new method proved to be useful. However,  in my opinion, the discussion could be improved by including some interesting data from other researches who were well successful on separation of underivatized metabolites from Cannabis using different HPLC methods in one unique step process on contrary to this new CG-MS pre-extraction method. There's many interesting work that also used RP-HPLC and HILIC columns to efficiently separate acid CDBs in fast way. So, it would be interesting to talk with those data in the discussion since it seems to be very practical method too.

         Our response:  We have improved the discussion and interpretation of the presented results. Moreover, we presented other liquid chromatography techniques (like. UPLC, HILIC) useful for the determination of cannabinoids:             see lines: 326-368; moreover, we add new references: [32 – 38].

Reviewer 4 Report

Comments and Suggestions for Authors

Author Response

Reviewer #4

 The aim of the work was to develop sensitive, selective and simple chromatographic methods for the

quantitative determination of trace CBD concentrations. As shown by the data presented, the

previously developed HPLC-DAD method does not provide adequate determination (high LOD and

LOQ) as well as separation from endogenous components in biological samples. Therefore, the

authors decided to develop a GC-MS method that would not have these disadvantages. It was also

assumed that the samples would not require a derivatization process . In my opinion, the goal has

not been fully achieved. Samples for GC analyses require an additional extraction step with NaOH

and another extraction with n-hexane, which lengthens the preparation process compared to HPLCDAD

analyses. In order to determine CBD-N and CBD-A concentrations, it is necessary to analyse 2

samples - supernatant N and layer A, each analysis takes more than 90 min, which again significantly

increases the time compared to the HPLC-DAD method (70 min analysis +18 min rinse), which allows

the determination of CBDs and CBDs-A in a single analysis. Although in both cases it is difficult to talk

about a simple and fast method, the GC-MS method provides better sensitivity and selectivity - and

only in this part the assumed research goal was met.

Our response:    The most important scientific novelty of our current study is the use of capillary GC-MS for quantification of underivatized CBDs-N, CBDs-A and Chol in biological materials. This goal was achieved by using simple CBDs-A extraction with 0.1M NaOH. Moreover, we avoid problems of determining the yield of the derivatization of CBDs possessing the large diversity of chemical structures; currently over 140 types of CBDs have been identified in plants (especially hemp or hemp seeds).

     Moreover, pre-column CBDs-A extraction with NaOH solution from processed samples reduces a number of analysed CBDs (i.e., without extracted CBDs-A) compared to GC-MS methods including pre-column derivatisation of CBDs. Therefore, we argued that elimination of CBDs derivatisation significantly improves the precision and accuracy of our original GC-MS method compared to previous GC-MS methods with pre-column derivatisation procedures as well as liqud chromatographic methods.

The use of IS, both in GC and HPLC, is questionable. GCIS is 5-a-cholestane, added to each sample in

the amount of 50μl of a solution with a concentration of 1.8 mg/ml n-hexane, and HPLCIS - 20μl of 2,5-

dimethylphenol solution with a concentration of 2.8 mg/ml methanol. This means that 90μg of GCIS

or 56μg of HPLCIS is introduced into each sample. This is a very high concentration compared to the

range of determinations of 2-30pg (according to Table 1) or 0.05-40 μg (Table 4), whether the

properties of this substance are similar to the tested CBDs, so that it is possible to conclude on the

basis of IS recovery CBDs extraction efficiency? Are the marking ranges given in tab. 1 and 4 are the

same for each of the tested substances? Chol assays appear to be performed in a different range, for

example as LOD and LOQ for GC-MS method are ~63 and 209 pg/ml vs. range 2-30pg (according to

table 1).

Our response:    Thank very much for your very valuable comments; indeed, the concentration of GC-IS is

  160 μg of 5-α-cholestane/mL of n-hexane (see line:  393).

                            The concentrations of CBDs and Chol in hexane (for GC-analyses) and methanol (for HPLC-DAD analyses) was the same; only injection volumes were different (usually 1-3 microL for GC-MS, whereas  1- 30 microL for HPLC-DAD analyses).   

                            The amount ranges of GC-MS injected standards 152 of CBDs and Chol: 2-30 and 3-39 pg/injection, respectively (Table 1).                        

                            The amount range of HPLC-injected CBDs and Chol standards dissolved in methanol: 0.05-0.40 microg/injection (see Table 4).

Was Chol also determined in samples of plant material?

 Our response:     No, the Chol- concentrations in plant samples were below LOD  (see Figs. 1B and 5B; especially

                              description under the Figures 1 and 5).

                               Indeed, Chol concetrations in plants are very low [39: Sonawane et al. Plant cholesterol biosynthetic pathway overlaps with phytosterol metabolism. Nat. Plants 2017, 3, 16205. https://doi.org/10.1038/nplants.2016.205].

Comparing Tables 1 and 4, where the parameters of both methods are listed, it can be concluded that the GC-MS method is at least an order of magnitude more sensitive (ng vs. μg for the calibration equation). The LOD and LOQ in the tables would be easier to compare if they were presented in comparable units (pg/ml vs ng/ml, and pmol/ml in brackets).

Our response: Appropriate corrections have been made; see Table 1 and 4.

In my opinion, the responses of MS and DAD detectors to 1 pg of the tested substance are in no way comparable, because they are completely different values (peak area in the case of MS and absorbance in the case of DAD), the last

column in the tables should be deleted.

Our response:   Please forgive me for leaving these last columns in tables 1 and 4.  The "responses" values of the detectors compare the detection efficiency of Chol and individual CBDs in the GC-MS system and compare the detection efficiency of Chol and individual CBDs in the HPLC-DAD system. Moreover, we suggested that these “response values/pg”  determine the suitability of MS-detector and DAD-detector in chromatographic analyzes of CBDs and Chol.

 The poor response of the detector eliminates its use in qualitative and especially quantitative analyses.

These tables require unification, e.g. arranging substances according to RT, inclusion of the RSD value in table 4, etc..

Our response:    We arranged substances according to RT (see Tables 1 and 4).

What is the point of calibration performed for IS if as a standard it is added to samples and standards in the same amount (50μl GCIS and 20μl HPLCIS). Additionally, information for HPLCIS is included in the description of Table 4, not in the table content.

Our response:    The concentration range of GCIS solutions: 2–10 µg GCIS/mL (see Table 1; footnote “l”);

                             The concentration range of HPLCIS solutions: 20 – 60 µg HPLCIS/mL (see Table 4; footnote “a”).

Table 1 is incomprehensible in the part regarding CBDs-A, why is the data for one substance placed in

two rows?

Our response:    The bottom row shows the retention time and Sn for the product (i.e., CBDs-N) of decarboxylation of CBDs-A  (see Table 1; footnote “k”)

In Table 4, LOD and LOQ are given in the header as [picomol/ml]? and in the description below the

table as [nmol/ml]

Our response:      This error has been removed;    see Table 4.

Tables and figures require standardization of descriptions. Information on how R, RSD, TF was

calculated should only be included in section 3.3.3. and do not repeat for each table. Some

information, such as explanations of the abbreviations CBDs, Chol, GCIS, SD, etc., which apply to all

tables, can only be included in the 1st table, and be referred to in the remaining tables.

Our response:       Information on how R, RSD,% and TF were calculated have been presented in section 3.3.3 (see Materials and Methods; see lines:  518 – 532.

In the footnotes below the tables 1-4 we have included only specific information about R, RSD,% or TF.

 In the chromatograms, adopt the same description of the peaks, i.e. 1 - CBDV, 1A - CBDV-A, 2 - THCV, 2A -

THCV-A, etc. In Fig. 1B instead of CBDV + CBDV-A there should be 1 + 1A etc.

Our response:       Appropriate corrections have been made (see Fig. 1A and 1B).

Figure 1 requires standardization of not only peak descriptions but also equal analysis time. In Fig. 1B

shows a peak at RT ~19min, which is not present in Fig. 1A, there is no description of this peak, while

in Fig. 1A shows the GCIS and Chol peaks that were missing in Fig. 1B.

Is it in Fig. 1 and 3 chromatograms are presented in scan or SIM mode?

Our response:       Appropriate corrections have been made (see Figs. IA and IB); we add information on TIC mode (for Figs. 1 and 3); see line 166 and lines: 448-449.

Detailed notes:

Chapter 2.1 The lack of problems with overlapping of the CBD, GCIS and Chol peaks in the standard

solution (Fig. 1A) and the tested biological sample (Fig. 1B), as well as interference from endogenous

components present in the tested biological materials, is probably due to the operation of the

detector in the SIM mode , which, however, is not described in "Results and discussion".

The reaction equation (Figure 2) was written incorrectly; placing CO2 above the arrow suggests that it

is one of the reactants, not the product.

Our response:  We used very shallow GC-MS-temperature programs A; therefore, we did not find any problems related to interference from endogenous components in the tested biological materials. Obvious, we used TIC-mode.

  1. Materials and methods.

Stock solutions (Chol, GCIS, and CBD standards) were prepared in 1 mL n-hexane.

Is the correct solvent given, are the ranges of all substances determined by a given method

the same? According to the descriptions in the tables, the concentration range in the GC-MS method

(Table 1) was 2-30 pg (? pg/ml or pg/injection), and for LC (Table 4) 0.05-0.40 μg (? μg/ml or

μg/injection). If the given amounts refer to an injection, different calibration curve ranges are

obtained due to different amounts of the injected sample. For uniformity, the range of calibration

curves can be given as concentrations (pg/ml or μg/ml).

Our response:  Thank you for your advice. We provide corrected data in Tables 1 and 4 (please see footnotes).

Was for GC analyses used He of purity 99.9999%, i.e. He 6.0, and not the more commonly used He

5.0 (99.999%)?

Our response:   We used He of purity: 99.9992% (see line: 390).

3.2.1. Hen egg yolk. The sample was prepared by extraction with n-hexane 4 times, and the residue

was dissolved in 1 mL of methanol before analysis by both GC-MS and C18-HPLC-DAD. However,

standards are prepared in hexane, also in chap. 3.3.1.1, we are talking about hexane samples. Was

the dry extraction residue dissolved in methanol for HPLC analyses and in n-hexane for GC, or was n-hexane

used in both cases?

 Our response:     Just before chromatographic analyses (i.e. GC-MS and HPLC), the residue was re-dissolved in 1 mL of GC-grade hexane (for GC-MS analyses) or 1 mL of HPLC-grade methanol (for C18-HPLC- DAD analyses); see lines: 412-415.

3.2.2. Plant materials. The powdered sample was extracted 2 times with methanol and 2 times with

hexane.

Why was a different extraction procedure used than for the Hen egg yolk samples.

Our response:     We used only n-hexane for pre-column preparation of egg-yolk samples, because egg-yolk is soluble

in methanol.

Were the samples of plant material dissolved in 1 ml of methanol (question as above) analysed only by GC-MS,

as stated in the text?

Our response:     Just before chromatographic analyses (i.e. GC-MS and  HPLC-DAD), the residue was re-dissolved in 1 mL of GC-grade hexane (for GC-MS analyses) or 1 mL of HPLC-grade methanol (for C18-HPLC-DAD analyses); see lines: 433 – 435.

3.3.1. Gas chromatographic analyses. Line 382-383 – it should be "The injector, transfer line and ion-source

temperatures..." please provide correct temperatures, e.g. the maximum temperature of the

ion source is 250°C, it cannot be 280°C.

Our response:      The injector and transfer line temperatures were maintained at 240oC and 330oC, respectively. The MS was operated in the EI mode and full scan monitoring (m/z 20–500); the ion-source temperature was set to 280°C (the ion-source limit of temperature: 300oC); see lines: 444-447.

      According to professional advice from THERMO-Serwis: The design of the MS system allows the temperature of the ion source to be set to a limit of 300°C, and the advantage is that the ITQ remains clean longer for analyzed samples, especially biological samples (before professional MS service maintenance by THERMO-Serwis).  You will find everything at this link on page 3 https://www.pragolab.sk/documents/ITQ.pdf .

Line 390 – final temperature is 234°C, while the previous isotherm was 320°C – please correct.

According to the given operating conditions of the chromatograph oven, the entire analysis takes

over 90 minutes. For what purpose, since the last Chol peak appears at 54.5 min, i.e. at a

temperature of ~300 C. Why does the analysis continue for 30-40 min with a further increase in

temperature? Were the patterns, plant and animal material analysed under the same conditions?

Our response:       Detailed GC-MS analyses of biological materials (particularly hemp or hen egg yolk samples) showed that complete removal of all endogenous components of assayed biological samples required raising the GC-column temperature to 334oC;  see lines 113-117.

 See column temperature program A; the chapter: 3.3.1 Gas chromatographic analyses.

Lines 395-396 – “total ion current (TIC) chromatograms and/or selected-ion monitoring (SIM)” –

nowhere is it stated which SIM characteristic ions were selected for the tested substances.

Our response:  Thank you for your correction. Indeed, we used only TIC-mode in our GC-MS analyses; see lines: 448– 449.  

3.3.2. Reversed-phase liquid-chromatographic (C18-HPLC-DAD) analyses. The work [30] mentioned in

the description of the C18-HPLC-DAD method does not seem to be the best source, other works by the same authors [27, 29] concern similar determinations and better illustrate the method used

Our response:      Indeed, we work [30] removed;  we add appropriate references: [31,40]:

 [31] Czauderna et al.  Simple HPLC analysis of tocopherols and cholesterol from specimens of animal origin. Chem. Anal. (Warsaw) 2009, 54, 203-214.

[40]  Czauderna  et al. Optimization of high-efficient pre-column sample treatments and C18-UFLC method for selective quantification of selected chemical forms of tocopherol and tocotrienol in diverse foods. Food Chem. 2024 437: 137909. https://doi.org/10.1016/j.foodchem.2023.137909

Round 2

Reviewer 4 Report

Comments and Suggestions for Authors

Author Response

DEAR REVIEVER: thank you very much for your valuable and helpful comments

  Reviewer 4  -  Comments and Suggestions: 

    The idea of using an internal standard (in simple terms) in analyses is to check the correctness of extraction, recovery value, repeatability of injection, etc. For this purpose, substances with the same or similar properties as the tested substances are used as an internal standard (e.g. isotopically labelled standards). At the same time, IS must not interfere with the analysis of test substances. To check the recovery and repeatability of the injection, IS is added in equal amounts to each sample. The IS concentration level should be similar to the substances being determined.

The ISs (5-a-cholestane or 2,5-dimethylphenol) used are not substances with properties similar to the tested CBDs and Chol. Moreover, their concentrations are too high compared to the substances tested.

Our response:  

        Large background fluctuations and the presence of endogenous substances in analysed biological materials (especially in egg yolk samples) were the reason for adding 20 µL of the stock HPLCIS (2,5-dimethylphenol) solution (i.e., 56 µg HPLCIS) to processed biological samples (see:  lines 312-315; Table 4, lines 334-335; Chapter 3.1 Standards and reagents, lines 405-410).  So, the amount of HPLCIS in these biological samples (final volume these samples: 1mL) injected onto C18-columns was 1.12 ng HPLCIS/injection; volume injection: 20 µL); thus, this amount of injected HPLCIS is similar to amounts of analysed CBDs and Chol (see Table 4, the footnote “b“:  “amount range of injected calibration CBDs and Chol standards dissolved in methanol: 0.05-0.40 mg/injection.“).  

       Moreover, as can be seen in TABLE 4, photodiode array detector (DAD) response to 1 pg of HPLCIS (2,5-dimethylphenol)  is poor (i.e. 0.171 ) compared with response to 1 pg of  CBDs or Chol. Considering the above, better precision or accuracy offered of higher amount of HPLCIS added to processed biological samples.  Thus, we obtain more precise and accurate information about the yield of the pre-column method.                     

           Similarly, IS as 5-α-cholestane is often used as the internal standard in GC-MS/FID analyses: “Previous studies showed that 5-α-cholestane (as the internal standard) has been used to quantify Chol, their oxidation product, steroids, fucosterol, sitosterols, campesterol or stigmasterol [27,40,41]. Unfortunately, the use of isotopically labelled (e.g. deuterium or 18O) Chol or selected CBDs (as internal standards) is very expensive.”; see lines: 370-375.   Please see also cited references:

  [14] Czauderna et al.”The sensitive and simple measurement of underivatized cholesterol and its oxygen derivatives in biological materials by capillary gas-chromatography coupled to a mass-selective detector.” Acta Chromatogr. 2013, 25(4), 655-667. https://doi: 10.1556/AChrom.25.2013.4.5;

  [40] Czerwonka et al. “Novel method for the determination of squalene, cholesterol and their oxidation products in food of animal origin by GC-TOF/MS. Int. J. Mol. Sci. 2024, 25, 2807. https://doi.org/10.3390/ijms25052807;

  [41] Hammond, E.W. Food and nutritional analysis | Oils and Fats. Editos: Paul Worsfold, Alan Townshend, Colin Poole, Encyclopedia of Analytical Science (Second Edition), Elsevier, 2005, pp. 328-334. ISBN 9780123693976. https://doi.org/10.1016/B0-12-369397-7/00186-2. (https://www.sciencedirect.com/science/article/pii/B0123693977001862) .

  The stock solution of GCIS contains  160 µg of 5-α-cholestane in mL n-hexane (see lines: 405-406). Next, to the processed biological samples, 50 µL of GCIS stock solution (i.e., 8 µg of GCIS) were added. The final volume of processed biological samples was 1 mL.  1 µl of these processed biological samples was   injected onto capillary GC-column.  See lines 406-410 ; Table 1 see lines 156-158.           Moreover, calibration GCIS solution (8 µg GCIS in 1 mL n-hexane) injected onto capillary GC-column - gives the peak-area = 299184 ; see Table 1 and the footnote “m”. Thus, the GCIS-peak area is similar to peak-areas (Sn) analysed CBDs and Chol (see Table 1). Furthermore: the amount of the calibration GCIS solution (8 µg GCIS/mL) is 8 ng/injection (Table 1, the footnote “m”); similarly: the amount ranges of injected CBDs and Chol standards: 2-30 and 3-39 ng/injection, respectively (Table 1, the footnote “d”).

      Due to the wide variety of CBDs chemical structures (144 types of CBDs), it is difficult to find an ideal internal standard. 

      Used GCIS (5-α-cholestane) and HPLCIS (2,5-dimethylphenol)  possess carbon-cyclic ring, methyl-groups, straight-chain alkanes; similarly, CBDs and cholesterol also possess carbon-cyclic ring, methyl-groups, straight-chain alkanes.  These chemical-groups stimulate the solubility of GCIS and HPLCIS in n-hexane and methanol (i.e., in organic solvents used for pre-column extraction of CBDs and Chol). We can therefore assume that the efficiency of the four extractions (with methanol and/or n-hexane) will be similar for GCIS, HPLCIS, CBDs and Chol.     Therefore, our original pre-column methods used four extractions, very effective a shaker (800 motion/min) and ultrasonication for 30-40 min. Moreover, before pre-column processing, analysed biological samples must be carefully crushed (i.e., finely powdered samples) or homogenized.  

        Indeed  (see lines 568-573):   “Detailed GC-MS and HPLC analyses showed that four extractions with n-hexane and/or methanol (see Chapter 3.2 Pre-column preparation procedures for biological materials) allowed for ≥96% CBDs and Chol extraction efficiency from analysed biological samples.

        The use of GCIS or HPLCIS is particularly important in routine analyses of many biological samples (especially plant and animals tissues rich in endogenous components, proteins or lipoproteins). In fact, GCIS and HPLCIS determine the effectiveness of separating supernatants (i.e., the solvents used for extractions of CBDs and Chol) from a bottom layer of processed biological samples (i.e., residues derived from analysed biological samples).” 

  Reviewer 4:   Please consider whether to delete data regarding IS use, due to its low usefulness.

     Our response:       The use of GCIS or HPLCIS is particularly important in routine analyses of many biological samples (especially plant and animals tissues rich in endogenous components, proteins or/and lipoproteins). In fact, GCIS and HPLCIS determine the effectiveness of separating supernatants (i.e., the solvents used for extractions of CBDs and Chol) from a bottom layer of processed biological samples (i.e., residues derived from analysed biological samples).” 

Reviewer 4:     For what purpose was the GCIS and HPLCIS calibration performed in the given concentration ranges (2-10 µg/ml and 20-60 µg/ml, respectively)?

Our response:    Number of points used in the calibration curves for the GCIS and HPLCIS was 5 (see Table 1, the footnote “d”) and 6 (see Table 4, the footnote “b”), respectively.

   We used these concentration ranges (i.e., 2-10 µg GCIS/ml and 20-60 µg HPLCIS/ml) because:  “To assayed biological samples, 50 µL of the stock solution of GCIS (i.e., 8 µg of GCIS) or 20 µL of the stock solution of HPLCIS (i.e., 56 µg of HPLCIS) were added. The final volume of processed biological samples injected onto GC- or HPLC-columns was 1 mL.”; see lines: 156-158 (Table 1), 334-335 (Table 4), 402-407, 424-427 and 445-447.

   Therefore, the amount range of GCIS in 1mL of calibration GCIS solutions was: 2–10 µg GCIS (the injection volumes: 1 µL; i.e., the amount range of injected GCIS onto the GC-column: 2-10 ng/injection), whereas the amount range of  HPLCIS was: 20-60 µg HPLCIS in 1 mL of calibration HPLCIS solutions (i.e., the amount range of injected HPLCIS onto the C18-columns: 0.4-1.2 mg/injection); see Table 1 (the footnote “l”) and Table 4 (the footnote “a”).

    Thus, these calibration solutions of GCIS and HPLCIS were used for preparing the calibration equations (i.e., the GCIS and HPLCIS calibrations). We suspect that the yield of the pre-column procedures was ≤ 100% . Thus, the lowest concentrations of GCIS and HPLCIS were 2 µg GCIS in 1 mL of GCIS solution and 20 µg HPLCIS in 1 mL of HPLCIS solution.   

 Reviewer 4:   Please correct the error in the text:

Reviewer 4:   Line 387 - "KOH were purchased from Merck" should be NaOH

Our response:    We replaced KOH with NaOH;  see line 399.

 Reviewer 4:   Line 446 – “m/z 20-500” is given, and line 441-2 – “the m/z range from 15 to 450

Our response:    We replaced “from 15 to 450”  with “from 20 to 500”;  see line 455.

Reviewer 4:     In the chapter Conclusion - line 546-548 "We recommend the use of pre-column extraction with NaOH solution, longer capillary GC-columns (e.g, 60 m) and GC-MS/MS techniques..." is unproven. The tests were performed on a 30 m column and using GC-MS operating in scan mode.

Our response:   We have corrected this statement:  For future studies, we suggest the implementation of pre-column extraction with NaOH solution, longer capillary GC columns (e.g. 60 m) and GC-MS/MS techniques for the determination of underivatised CBDs, Chol and oxidised Chol (Ox-Chol) in animal tissues (especially Chol‑rich) and underivatised CBDs and Chol in plant materials like hemp).”; see lines 561-565
